# Inflammation-Induced Protein Unfolding in Airway Smooth Muscle Triggers a Homeostatic Response in Mitochondria

**DOI:** 10.3390/ijms22010363

**Published:** 2020-12-31

**Authors:** Debanjali Dasgupta, Philippe Delmotte, Gary C. Sieck

**Affiliations:** Department of Physiology & Biomedical Engineering, Mayo Clinic, Rochester, MN 55905, USA; Dasgupta.debanjali@mayo.edu (D.D.); Delmotte.Philippe@Mayo.edu (P.D.)

**Keywords:** reactive oxygen species (ROS), oxidative stress, Mfn2, Drp1, MCU, asthma

## Abstract

The effects of airway inflammation on airway smooth muscle (ASM) are mediated by pro-inflammatory cytokines such as tumor necrosis factor alpha (TNFα). In this review article, we will provide a unifying hypothesis for a homeostatic response to airway inflammation that mitigates oxidative stress and thereby provides resilience to ASM. Previous studies have shown that acute exposure to TNFα increases ASM force generation in response to muscarinic stimulation (hyper-reactivity) resulting in increased ATP consumption and increased tension cost. To meet this increased energetic demand, mitochondrial O_2_ consumption and oxidative phosphorylation increases but at the cost of increased reactive oxygen species (ROS) production (oxidative stress). TNFα-induced oxidative stress results in the accumulation of unfolded proteins in the endoplasmic reticulum (ER) and mitochondria of ASM. In the ER, TNFα selectively phosphorylates inositol-requiring enzyme 1 alpha (pIRE1α) triggering downstream splicing of the transcription factor X-box binding protein 1 (XBP1s); thus, activating the pIRE1α/XBP1s ER stress pathway. Protein unfolding in mitochondria also triggers an unfolded protein response (^mt^UPR). In our conceptual framework, we hypothesize that activation of these pathways is homeostatically directed towards mitochondrial remodeling via an increase in peroxisome proliferator-activated receptor-gamma coactivator 1 alpha (PGC1α) expression, which in turn triggers: (1) mitochondrial fragmentation (increased dynamin-related protein-1 (Drp1) and reduced mitofusin-2 (Mfn2) expression) and mitophagy (activation of the Phosphatase and tensin homolog (PTEN)-induced putative kinase 1 (PINK1)/Parkin mitophagy pathway) to improve mitochondrial quality; (2) reduced Mfn2 also results in a disruption of mitochondrial tethering to the ER and reduced mitochondrial Ca^2+^ influx; and (3) mitochondrial biogenesis and increased mitochondrial volume density. The homeostatic remodeling of mitochondria results in more efficient O_2_ consumption and oxidative phosphorylation and reduced ROS formation by individual mitochondrion, while still meeting the increased ATP demand. Thus, the energetic load of hyper-reactivity is shared across the mitochondrial pool within ASM cells.

## 1. Introduction

Airway inflammation underlies a number of pathological conditions such as asthma, chronic obstructive pulmonary disease (COPD), chronic bronchitis, COVID-19 and various cough syndromes. With the COVID-19 pandemic, we are keenly aware of the detrimental impact of airway inflammation and acute respiratory distress syndrome (ARDS). Even before COVID-19, ARDS affected as many as 246,000 people in the USA each year (based on an incidence rate of ~75 per 100,000), with influenza and the common cold affecting many more. The effects of acute airway inflammation are mediated by pro-inflammatory cytokines (e.g., tumor necrosis factor alpha, TNFα) leading to increased contractile protein expression, increased force generation and increased ATP consumption in airway smooth muscle (ASM) [1,2]. In humans, the increase in ATP consumption in ASM is matched by increased mitochondrial O_2_ consumption at the expense of increased reactive oxygen species (ROS) formation (oxidative stress) [3]. The subsequent oxidative stress results in an accumulation of unfolded proteins in the endoplasmic reticulum (ER) and in mitochondria triggering unfolded protein responses in both organelles. In a recently published study, we demonstrated that TNFα selectively activates the inositol-requiring enzyme 1 alpha (pIRE1α) mediated ER stress pathway in human ASM, which induces alternative splicing of mRNA for the transcription factor X-box binding protein 1 (XBP1s) [3]. In a follow-up study, we showed that 24 h TNFα exposure increases the expression of peroxisome proliferator-activated receptor-gamma coactivator 1 alpha (PGC1α) in human ASM cells, thereby promoting mitochondrial biogenesis and an increase in mitochondrial volume density [4]. The TNFα induced increase in PGC1α and mitochondrial remodeling could be downstream to either ER stress or the mitochondrial unfolded protein response (^mt^UPR) [5,6]. In addition to playing a key role in mitochondrial biogenesis, PGC1α also activates the Phosphatase and tensin homolog (PTEN)-induced putative kinase 1 (PINK1)/Parkin mitophagy pathway, which is involved in the ubiquitination of the mitochondrial fusion protein Mfn2 [7,8]. Accordingly, we found that acute exposure to TNFα also results in a decrease in mitofusin 2 (Mfn2) protein levels in human ASM cells while expression of dynamin-related protein-1 (Drp1) increases, resulting in mitochondrial fragmentation [4,9,10]. In addition to promoting mitochondrial fusion, Mfn2 also plays an essential role in tethering mitochondria to the ER, thereby establishing the proximity of mitochondria to ER Ca^2+^ release sites that represent microdomains of higher cytosolic Ca^2+^ concentrations ([Ca^2+^]_cyt_ “hotspots”) [9,10,11,12]. In another study, we reported that acute TNFα exposure disrupts the tethering of mitochondria to the ER and reduced mitochondrial Ca^2+^ influx through the mitochondrial Ca^2+^ uniporter (MCU) [13]. This is consistent with other studies showing that close proximity of mitochondria to the ER is necessary to open the MCU and thereby promote mitochondrial Ca^2+^ influx [9,10,11,12]. In this review article, we provide a conceptual framework for a homeostatic response triggered by activation of the pIRE1α/XBP1s ER stress pathway, which serves to mitigate the impact of TNFα-induced ER stress by promoting mitochondrial biogenesis, reducing mitochondrial Ca^2+^ influx and thereby limiting ROS formation (Figure 1). Components of this homeostatic response include a pIRE1α/XBP1s-mediated increase in PGC1α and an increase in PINK1/Parkin leading to: (a) a reduction in Mfn2 and restricted tethering of mitochondria to the ER, and (b) mitochondrial biogenesis leading to an increase in mitochondrial volume density, which reduces O_2_ consumption and ROS formation per mitochondrion, while still meet the increased ATP demand of force generation.

## 2. TNFα Increases ASM Force Generation, ATP Consumption and Tension Cost

In ASM cells, agonist (e.g., acetylcholine—ACh)-induced increase in [Ca^2+^]_cyt_ leads to a contractile response through a signaling cascade that involves Ca^2+^ binding to calmodulin (CaM), Ca^2+^/CaM mediated activation of myosin light chain kinase (MLCK), phosphorylation of the regulatory myosin light chain (rMLC_20_), myosin heavy chain (MyHC) binding to actin (cross-bridge formation), attachment of actin filaments to the ASM membrane and force generation against an external load [2,14,15]. ASM force generation resulting from cross-bridge recruitment and cycling is driven by ATP hydrolysis [1,2,16,17] (Figure 2). Recently, we showed that TNFα increases ASM force generation in response to muscarinic (ACh) stimulation [1,2]. The TNFα-induced increase in ASM force is primarily due to an increase in contractile protein expression and is accompanied by a marked increase in ATP consumption and tension cost (ATP hydrolysis rate per unit of isometric force) [1,2]. Under normal physiological conditions, the tension cost of ASM is significantly lower than skeletal muscle; however, work efficiency is higher [16]. Thus, the energetics of ASM are perfectly suited to sustain a greater force at a lower energy cost.

## 3. TNFα Induced Oxidative Stress Triggers Protein Unfolding in the ER and Mitochondria

In human ASM, we previously showed that TNFα exposure and increased ROS formation lead to an accumulation of unfolded proteins in the ER lumen, which trigger a homeostatic signaling cascade to restore normal function (i.e., ER stress response) [3]. As a positive control, we showed that all three ER stress signaling pathways (protein kinase RNA-like ER kinase (PERK), activating transcription factor 6 (ATF6) and IRE1α) were activated in response to tunicamycin. However, in response to TNFα, only the pIRE1α/XBP1s ER stress pathway was selectively activated in human ASM cells. Importantly, although the PERK and ATF6 ER stress pathways were activated by tunicamycin, they were not activated by TNFα [3]. The involvement of TNFα-induced ROS formation in triggering the ER stress response in human ASM was confirmed by the fact that the ROS scavenger tempol blunted TNFα-induced activation of the pIRE1α/XBP1s ER stress pathway [3]. Moreover, we found that TNFα did not induce changes in expression of proteins downstream from activation of the PERK and ATF6 ER stress pathways (e.g., phosphorylation of eukaryotic initiation factor 2 alpha (p-eIF2α), activating transcription factor 4 (ATF4) and CCAAT-enhancer-binding protein homologous protein (CHOP)). Selective activation of ER stress pathways is not restricted to the pIRE1α/XBP1s ER stress pathway but can involve PERK or ATF6 pathways as well [3,18]. It is possible that selective activation of ER stress pathways depends on the inducer and/or cell type. In this regard, it is possible that in ASM cells, the binding affinity of IRE1α to the Grp78/BiP protein is lower than other ER stress sensors. Grp78/BiP dissociation triggers autophosphorylation of IRE1α and activation of the pIRE1α /XBP1s ER stress pathway. In asthmatic airway epithelial cells, it was reported that the expression of ER stress markers is elevated [19,20,21]; however, activation of specific ER stress pathways was not examined (e.g., phosphorylation of IRE1α or PERK, or cleavage of ATF6). In a mouse model of asthma, chemical chaperones were shown to reduce the expression of ER stress markers in lung tissue and attenuate airway hyperresponsiveness [22,23,24,25]. However, as in the case of asthmatic airway epithelial cells, these studies using a mouse model did not examine the specific activation of different ER stress pathways.

Increased mitochondrial ROS formation can also lead to mitochondrial protein damage and unfolding, triggering the ^mt^UPR [5,26]. As with the ER stress response, the ^mt^UPR is an adaptive response to buffer excess ROS, refold proteins or remove damaged proteins. Although the mechanisms of the ^mt^UPR have been extensively studied in *Caenorhabditis elegans*, the sensory mechanism of ^mt^UPR in vertebrate tissue is not clear. The canonical ^mt^UPR pathway involves crosstalk between mitochondria and the nucleus via the recruitment of transcription factors such as CHOP, ATF4, and activating transcription factor 5 (ATF5). Acting together, these chaperones and transcription factors promote an integrated stress response (ISR) [27,28,29], resulting in the production of various antioxidants, chaperones and proteases to reduce stress and protein damage inside the mitochondrial matrix [5,6]. This ISR is coordinated with an ER stress response by phosphorylation of eIF2α as a downstream effector of the PERK/eIF2α ER stress pathway, which can be activated by increased ROS, amino acid depletion and ER dysfunction [26,30]. Considering the pro-apoptotic properties of CHOP and ATF5, it is hypothesized that uncontrollable ^mt^UPR can lead to cellular apoptosis [5,28,31]. However, as mentioned above, we found that TNFα exposure did not affect CHOP or ATF4 expression, nor does it trigger apoptosis. In addition to the canonical CHOP-ATF4-ATF5 mediated pathway, the ^mt^UPR also activates sirtuin-3-mediated transcriptional upregulation of superoxide dismutase 2 (SOD2) and catalase as an antioxidant response [5]. We cannot exclude a role of this ^mt^UPR pathway, particularly in mitigating mitochondrial ROS formation in ASM after TNFα exposure.

Another important consequence of mitochondrial ROS formation is transient opening of mitochondrial permeability transition pore (mPTP) and dissipation of the mitochondrial membrane potential (Δψ_m_) [32] (Figure 2). During conditions of excess ROS and Ca^2+^ overload, opening the mPTP helps to maintain Ca^2+^ levels in the mitochondrial matrix [33,34]. However, a dysregulation in mPTP opening results in the release of various matrix metabolites including NAD+ and leads to the Δψ_m_ depolarization, inhibition of oxidative phosphorylation and ultimately mitochondrial damage [35,36,37].

## 4. TNFα-Induced Activation of pIRE1α/XBP1s ER Stress Pathway Increases PGC1α Expression, Mediating Mitochondrial Fragmentation, and an Increase in Mitochondrial Biogenesis

In a recent study in human ASM, we found that TNFα-induced activation of the pIRE1α/XBP1s ER stress pathway is associated with an increase in PGC1α expression (Figure 3A) [4], a protein known to play a significant role in regulating mitochondrial biogenesis and dynamic remodeling [38,39,40]. Importantly, PGC1α is a known transcriptional target of XBP1s [41]. Additionally, PGC1α can modulate the activity of another protein, PTEN-induced putative kinase 1 (PINK1) by cooperative binding and helps in its activation [42,43]. PINK1, together with Parkin, forms a well-known regulatory complex of the mitophagy pathway involved in mitochondrial remodeling. In other cell types, previous studies have reported that PINK1 mediates the phosphorylation of Mfn2, which is an important step in Parkin-mediated ubiquitination of Mfn2 [7,8,43,44]. However, the cause-and-effect link between activation of the pIRE1α/XBP1s ER stress pathway and the downstream regulation of the PINK1/Parkin-mediated mitophagy pathway and reduced Mfn2 remains to be fully explored.

In previous studies, we also observed that TNFα induces mitochondrial fragmentation in human ASM cells, which is consistent with a decrease in the mitochondrial fusion protein Mfn2 and an increase in the mitochondrial fission protein Drp1 [4]. The TNFα-induced decrease in Mfn2 in ASM cells is dependent on activation of the pIRE1α/XBP1s ER stress pathway, and likely relates to the PINK1/Parkin mitophagy pathway, as mentioned above. Based on our conceptual framework (Figure 1), we hypothesize that mitochondrial remodeling reflects a coordinated homeostatic response triggered by inflammation and TNFα exposure in human ASM cells. Another component of this homeostatic response involves an increase in mitochondrial biogenesis and mitochondrial volume density (Figure 3B,C) to cope with the excess ATP demand. With an increase in mitochondrial volume density, O_2_ consumption and the production of ATP would be shared across more mitochondria, thereby reducing ROS production by individual mitochondrion, and diminishing their risk for oxidative injury and cell death.

Mitochondria undergo continuous fusion and fission, which are essential for optimal mitochondrial function [45]. Three major proteins (i.e., mitofusins 1/2 (Mfn1/Mfn2), optic atrophy 1 (OPA1), and Drp1) balance the equilibrium between mitochondrial fusion and fission (Figure 4A), maintaining the functional activity of mitochondria when cells experience metabolic or environmental stress [45,46,47]. In a previous study, we found that exposing human ASM cells to cigarette smoke extract causes mitochondrial fragmentation by reducing Mfn2 and increasing Drp1 [48]. Recently, we found that acute exposure to TNFα also induces a reduction in Mfn2 and an increase in Drp1 expression in human ASM cells (Figure 4B) with an associated mitochondrial fragmentation (Figure 4C,D) [4]. Mitochondria in human ASM cells were visualized by confocal imaging of MitoTracker labeling. Following 3D reconstruction, two specific mitochondrial morphological characteristics were calculated: (1) form factor, defined as the length of mitochondria^2^/4π* area of mitochondria (Figure 4C), and (2) aspect ratio, defined as the ratio of the major and minor axes of mitochondria (Figure 4D) [4]. Lower values of both form factor and aspect ratio indicate fragmentation (Figure 4C,D).

Previous studies have suggested that a reduction in Mfn2 and mitochondrial fragmentation are actually upstream of ER stress [49,50,51,52]. These studies in neurons, embryonic fibroblasts and Drosophila reported that Mfn2 knockdown activates the pIRE1α/XBP1s ER stress pathway [49,51,52]. Consequently, these studies also suggested that a reduction in Mfn2 can alter the morphology of ER, as well as the proximity of mitochondria to the ER, thereby triggering ER stress [51,53]. However, our results in human ASM clearly indicate that the TNFα-induced reduction in Mfn2 is downstream of the activation of the pIRE1α/XBP1s ER stress pathway [3]; however, this does not preclude a vicious cycle that would amplify oxidative stress and cell injury unless mitigated by a homeostatic response.

## 5. TNFα-Induced Reduction in Mfn2 Disrupts Mitochondrial Tethering to ER and Reduces Mitochondrial Ca^2+^ Influx

In ASM, the TNFα-induced increase in force and ATP consumption stimulates F1Fo-ATPase/ATP synthase (complex V) activity and increases O_2_ consumption and ATP production in mitochondria. Based on biochemical studies, it is well known that mitochondrial production of ATP (oxidative phosphorylation) also depends on dehydrogenase enzyme activities of the TCA cycle, particularly pyruvate dehydrogenase (PDH), NAD-isocitrate dehydrogenase (ICDH), and oxoglutarate dehydrogenase (OGDH), which are Ca^2+^ dependent [54,55,56,57,58]. Additionally, an increase in [Ca^2+^]_cyt_ increases NADH levels in the mitochondria by stimulating mitochondrial shuttle systems such as the glycerol phosphate shuttle and the aspartate/glutamate transporters [59,60,61]. Thus, mitochondrial Ca^2+^ influx during transient elevation of [Ca^2+^]_cyt_ stimulates dehydrogenase enzyme activities within the TCA cycle and increases O_2_ consumption, electron transport chain (ETC) flux and ATP production through excitation–energy coupling (Figure 2). Conversely, it is well known that increased ATP consumption leads to transport of ADP into mitochondria via the adenosine nucleotide transporter (ANT), which stimulates F1Fo-ATPase/ATP synthase (complex V) activity [62] to match ATP production with ATP consumption (Figure 2). Together, the excitation–contraction mediated ATP hydrolysis and subsequent production of ATP by ATP synthase ultimately forms a normal homeostatic mechanism that we termed excitation–energy coupling in a variety of cell types including human ASM [9,10,63]. Pathophysiology, such as airway inflammation and mitochondrial dysfunction, involves disruptions in these mitochondrial energy-sensing/signaling pathways. Furthermore, a consistent elevation in [Ca^2+^]_mit_ can trigger the opening of mPTP and ultimately can lead to mitochondrial dysfunction and cellular death [35,36]. In previous studies in ASM cells, we showed that the transient [Ca^2+^]_cyt_ response to 1 µM ACh stimulation is accompanied by a transient elevation of [Ca^2+^]_mito_ [9,13,64] and we confirmed that the transient [Ca^2+^]_mito_ response was inhibited by blocking the mitochondrial Ca^2+^ uniporter (MCU). Mitochondrial Ca^2+^ influx via the MCU is only activated by microdomains of higher [Ca^2+^]_cyt_ (“hotspots” >2 µM) [9,10,11,12], which exceed the normal global transient [Ca^2+^]_cyt_ response to agonist stimulation in ASM (~500–600 nM). Higher levels of [Ca^2+^]_cyt_ are present at the ER Ca^2+^ release sites (IP_3_ and RyR channels). Thus, during agonist stimulation, mitochondria must be tethered to the ER to provide close proximity to [Ca^2+^]_cyt_ “hotspots” in order to activate the MCU for mitochondrial Ca^2+^ influx. Although TNFα exposure increases the [Ca^2+^]_cyt_ response to ACh in ASM cells, the [Ca^2+^]_mito_ response is markedly reduced [64], reflecting a decrease in mitochondrial Ca^2+^ influx. Recent studies have shown that TNFα exposure reduces the motility of mitochondria and disrupts the tethering of mitochondria to ER, thereby affecting the proximity of mitochondria to [Ca^2+^]_cyt_ “hotspots” (Figure 5) [13]. This disruption of mitochondrial/ER proximity results in reduced MCU activation and a decrease in transient mitochondrial Ca^2+^ influx during agonist activation. The reduction in mitochondrial Ca^2+^ concentration ([Ca^2+^]_mito_) following acute TNFα exposure subsequently reduces TCA cycle activity, O_2_ consumption and ROS formation (Figure 2) [13,64]. TNFα exposure reduced proximity of mitochondria to the ER, which is also consistent with the uncoupling of transient [Ca^2+^]_cyt_ and [Ca^2+^]_mito_ responses to agonist stimulation due to an inability to establish hot spots for mitochondrial Ca^2+^ influx via the MCU.

The disruption of proximity of mitochondria in ASM cells induced by acute TNFα is likely due to the reduction in Mfn2. Mfn2 is located at the ER membrane and dimerizes with Mfn2 (Mfn2/Mfn2) and/or Mfn1 (Mfn1/Mfn2) located at the mitochondrial membrane, and thereby plays an essential role in tethering mitochondria to the ER [9,10,11,12,53,58,65]. In a published study from our lab [13], mitochondrial proximity to the ER was assessed using fluorescent labeling of the ER and mitochondria, confocal imaging and 3D reconstruction and overlap areas were estimated using the Manders’ overlap coefficient (Figure 5A). We found that acute TNFα exposure markedly reduced the proximity of mitochondria to the ER (Figure 5A) [13]. The proximity of mitochondria to the ER was also assessed using 3D EM (Figure 5B). Consistent with results from confocal imaging (Figure 4C,D) [4], we found that mitochondria in human ASM cells were fragmented after 24 h exposure to TNFα (Figure 5B) [13].

In addition, the mitochondrial proximity to the ER was reduced. In addition to Mfn2 mediated tethering, TNFα exposure also disrupts basal mitochondrial movement by reducing the expression of two mitochondrial motility associated proteins: miro (mitochondrial Rho GTPase protein, also called RhoT1/2) and milton (a trafficking protein kinesin, TRAK) [13]. These two proteins help in the coupling of mitochondria with microtubules and maintain the directed trajectory or kinetic, more random motion of mitochondria, which is necessary to establish a proximity to the ER (or Ca^2+^ hotspots) [66]. Thereby disrupting the ability of mitochondria to establish proximity to the ER, TNFα reduces the transient mitochondrial Ca^2+^ uptake during agonist activation and ultimately reduces the O_2_ consumption and oxidative phosphorylation for each mitochondrion.

## 6. TNFα-Induced Mitochondrial Biogenesis and Increased Volume Density Increases O_2_ Consumption and ATP Production While Mitigating ROS Formation

Previous studies have shown that TNFα exposure increases mitochondrial O_2_ consumption rate in ASM (Figure 6) [4]. However, TNFα also increases mitochondrial volume density in ASM cells (Figure 6A,B) [4]. As a result, when corrected for the increase in mitochondrial volume, O_2_ consumption rate per mitochondrion is reduced (Figure 6C,D) [4]. Underlying the TNFα-induced increase in mitochondrial volume density, we found that mitochondrial biogenesis was increased as reflected by an increase in DNA copy number (Figure 3B,C). Interestingly, studies have reported that mitochondrial biogenesis is also enhanced in asthmatic ASM [67,68]; however, these studies did not examine mitochondrial network morphology or the mechanisms mediating mitochondrial biogenesis. An increase in mitochondrial volume density is an alternative mechanism to increase overall cellular O_2_ consumption and ATP production to meet the increased ATP demand in human ASM after exposure to pro-inflammatory cytokines. However, as a result of the increase in mitochondrial volume, the O_2_ consumption per mitochondrion is reduced (Figure 6D), thereby minimizing formation of ROS per mitochondrion (Figure 6F). With the overall increase in O_2_ consumption in ASM cells induced by TNFα, ROS formation also increases (Figure 6E) [3]. It should be noted again that TNFα also increases mitochondrial biogenesis and volume density in human ASM (Figure 6A,B). Thus, when the TNFα-induced ROS formation was normalized for mitochondrial volume density, ROS formation per mitochondrion was mitigated (Figure 6F). This increase in mitochondrial volume is a possible homeostatic response to maintain the increased ATP production while reducing the impact of O_2_ consumption and ROS formation per mitochondrion, and overall scavenging (by SOD, etc.) would be more effective.

## 7. The Homeostatic Response to TNFα in Asthmatic Human ASM

Asthma is a chronic inflammatory condition of ASM, characterized by airway inflammation, airway hyper-contractility and airway remodeling (ASM proliferation) [69,70,71,72,73]. Surprisingly, very few studies on asthma have dealt with the crosstalk between mitochondrial oxidative stress and ER stress response and its impact on the inflammation in human ASM to date. An increased ROS generation is reported in asthmatic patients [69,74,75], which can trigger the ER stress in human ASM. Some studies have demonstrated that the activation of ER stress response in airway epithelial cells or immune cells is enhanced in asthma [19,20,21] but did not specifically explore the contribution of any of the ER stress sensor activation in this context. However, to date, no study has explored the impact of ER stress and its marker activation in human asthmatic ASM. As mentioned above, pro-inflammatory cytokine TNFα selectively induces the pIRE1α/XBP1s ER stress pathway in non-asthmatic ASM [3], and the effect of inflammation is not explored in asthmatic ASM. It is not known whether TNFα mediated ER stress plays a similar homeostatic role maintaining the dynamicity and activity of mitochondria in asthmatic conditions or if the effect is blunted. Additionally, the impact of any other proinflammatory cytokines other than TNFα in asthma is also a question to be answered. Previous studies have shown that in asthmatic ASM, mitochondrial biogenesis is increased [67,68], and we can speculate that it can be positively correlated with mitochondrial volume density, however, it has not been reported so far. We have reported an increase in mitochondrial fragmentation in human ASM of moderate asthmatics associated with an increase in mitochondrial fission protein Drp1 and a decrease in fusion protein Mfn2 expression in asthma, resulting in more fragmented mitochondrial networks [48]. Considering all the experimental evidence, further illustrating studies are needed to understand the impact of inflammation on ER stress and mitochondrial crosstalk in asthma and also to find some novel therapeutic target(s).

## 8. Conclusions

With the recent emergence of the COVID-19 pandemic, we became concerned with the pathological significance of acute airway inflammation, which is mediated by pro-inflammatory cytokines (e.g., TNFα). TNFα plays a major role in force generation and ATP consumption by ASM during airway inflammation and induces mitochondrial oxidative phosphorylation to match the increased ATP demand. As a result, ROS generation and oxidative stress is increased, which can be deleterious to the system. Simultaneously, as a homeostatic response, TNFα selectively activates the pIRE1α/XBP1s mediated ER stress pathway in ASM and increases mitochondrial biogenesis and mitochondrial volume density and reduces O_2_ consumption and ROS formation by individual mitochondrion and protects ASM cells from the negative impact of inflammation-induced ROS formation while still meeting the ATP demand. In our belief, a failure in this homeostatic mechanism can lead to increased ROS formation, thereby exacerbating cell death.

## Figures and Tables

**Figure 1 ijms-22-00363-f001:**
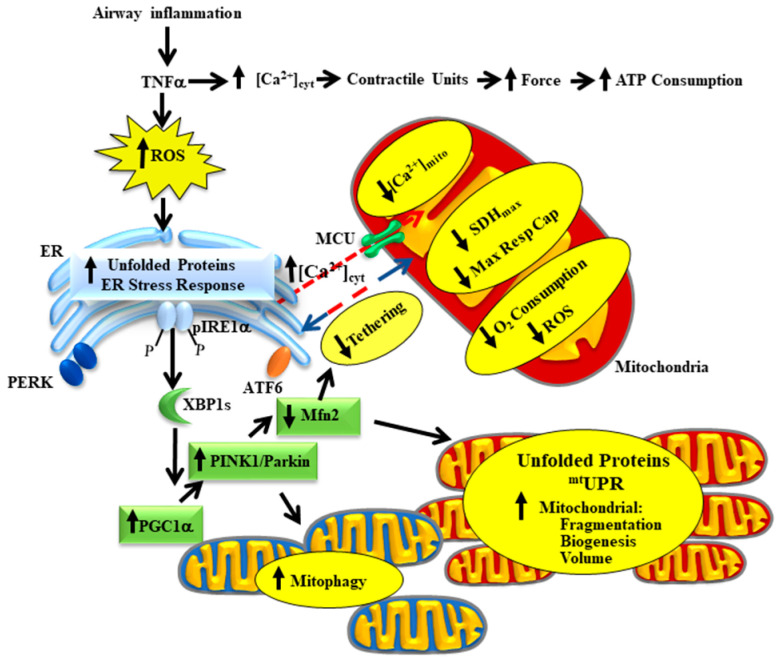
Our conceptual model hypothesizes that, following airway inflammation, TNFα induces an increase in airway smooth muscle (ASM) force generation and ATP consumption leading to increased mitochondrial O_2_ consumption and reactive oxygen species (ROS) production. TNFα-induced ROS formation leads to protein unfolding and selective activation of the pIRE1α/XBP1s endoplasmic reticulum (ER) stress pathway and a mitochondrial unfolded protein response (^mt^UPR). Downstream XBP1s mediates an increase in PGC1α, PINK1/Parkin expression resulting in increased Drp1 and reduced Mfn2 expression, which leads to mitochondrial fragmentation and mitophagy. Reduced Mfn2 also disrupts mitochondrial tethering to the ER, thereby decreasing mitochondrial Ca^2+^ influx. Increased PGC1α also promotes mitochondrial biogenesis and increased mitochondrial volume density. The homeostatic remodeling of mitochondria reduces O_2_ consumption and ROS formation by individual mitochondrion while still meeting the increased ATP demand. Thus, the energetic load of hyper-reactivity is shared across the mitochondrial pool within ASM cells.

**Figure 2 ijms-22-00363-f002:**
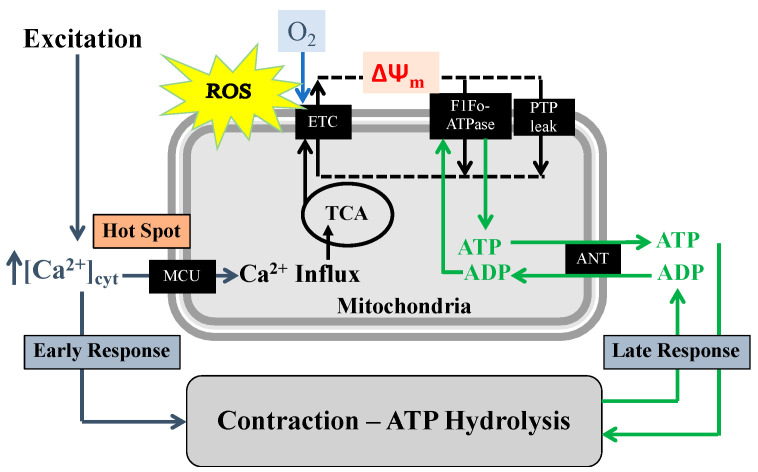
Our conceptual model demonstrates that in ASM cells, excitation, contraction, and energy production are coupled with two mechanisms. (1) On a shorter timescale, agonist-induced elevation of [Ca^2+^]_cyt_ triggers activation of the mitochondrial Ca^2+^ uniporter (MCU) and Ca^2+^ influx, which increases tricarboxylic acid cycle (TCA) activity thereby providing H^+^ to the electron transport chain (ETC). (2) On a longer timescale, agonist-induced force/contraction results in ATP consumption and an increase in ADP, which increases ADP influx into mitochondria via the adenosine nucleotide transporter (ANT), stimulating ATP synthase (F1Fo-ATPase) activity. With excessive mitochondrial Ca^2+^ influx, an overload in mitochondrial Ca^2+^ concentration ([Ca^2+^]_mit_) can trigger the opening of mitochondrial permeability transition pores (mPTP), which are large conductance channels present in the inner mitochondrial membrane that reduce the H^+^ gradient and mitochondrial membrane potential (Δψ_m_).

**Figure 3 ijms-22-00363-f003:**
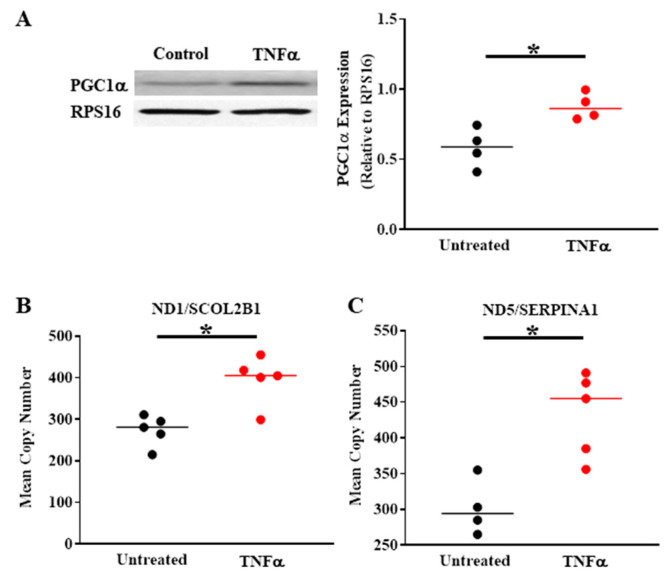
Compared to untreated controls, TNFα (20 ng/mL, 24 h) increases PGC1α protein expression in human ASM (**A**) and mitochondrial DNA copy number (**B**,**C**), indicating an increase in mitochondrial biogenesis. (* denotes a significant difference between TNFα treated and untreated control groups; *p* < 0.05; *t*-test; cells were dissociated from ASM samples from *n* = 5 patients and divided into TNFα treated and untreated control groups). All data are presented as scatter plots with the lines indicating the mean values. These figures are modified from presentations of previously reported results [4].

**Figure 4 ijms-22-00363-f004:**
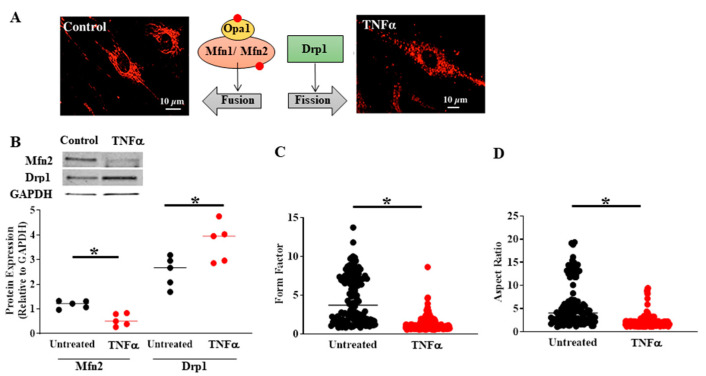
Compared to untreated controls, TNFα (20 ng/mL, 24 h) exposure induced mitochondrial fragmentation in human ASM (**A**) associated with reduced Mfn2 and increased Drp1 protein levels (**B**). The extent of mitochondrial fragmentation was assessed by calculating (**C**) form factor and (**D**) aspect ratio of mitochondria labeled using MitoTracker Red (**A**). In B, cells were dissociated from ASM samples from *n* = 5 patients and divided into TNFα treated and untreated control groups. In C and D, multiple mitochondria/cells were measured from ASM cells isolated from *n* = 5 patients divided into TNFα treated and untreated control groups. (* denotes a significant difference between TNFα treated and untreated control groups; *p* < 0.05; *t*-test). All data are presented as scatter plots with lines indicating mean values. These figures are modified from presentations of previously reported results [4].

**Figure 5 ijms-22-00363-f005:**
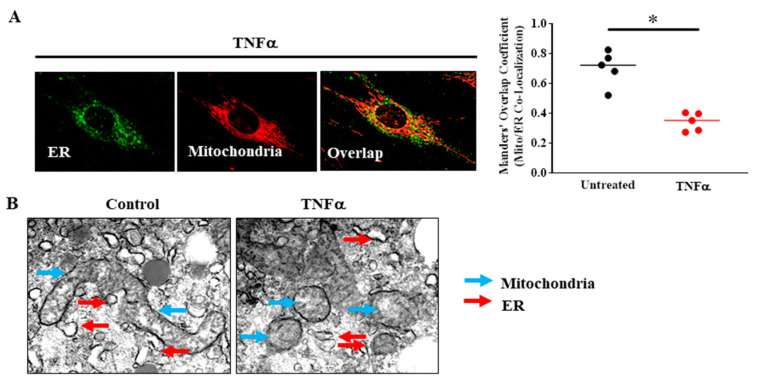
(**A**) In human ASM cells, ER was labeled using BODIPY FL thapsigargin (endoplasmic reticulum Ca^2+^ pumps, green) and mitochondria were labeled using MitoTracker red and imaged in 3D by confocal microscopy. Proximity of mitochondria to the ER was determined by measuring the Manders’ overlap co-efficient. Acute TNFα exposure (20 ng/mL, 24 h) reduced proximity of mitochondria to the ER (* denotes a significant difference between TNFα treated and untreated control groups; *p* < 0.05; *t*-test; cells were dissociated from ASM samples from *n* = 5 patients and divided into TNFα treated and untreated control groups). (**B**) 3D EM images showing that mitochondria (blue arrows) in human ASM cells are fragmented after TNFα (20 ng/mL, 24 h) exposure and that the incidence of close proximity of mitochondria to ER (red arrows) was reduced (cells were dissociated from ASM samples from *n* = 2 patients and divided into TNFα treated and untreated control groups). All data are presented as scatter plots with lines indicating the mean. These figures are modified from presentations of previously reported results [13].

**Figure 6 ijms-22-00363-f006:**
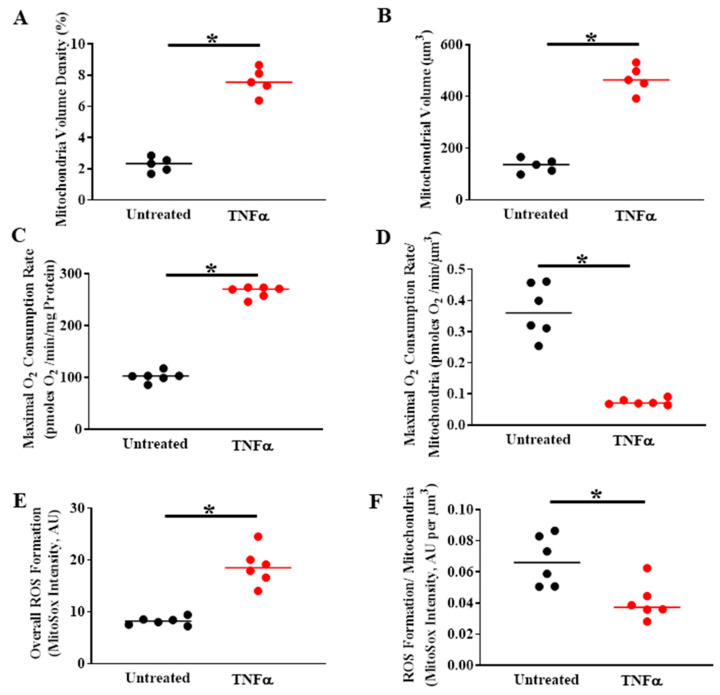
In human ASM cells, TNFα exposure (20 ng/mL, 24 h) increases (**A**) mitochondrial volume density per cell and (**B**) and the total mitochondrial volume and (**C**) overall maximum mitochondrial O_2_ consumption rate (measured using Seahorse) (**D**) maximum O_2_ consumption rate when normalized for mitochondrial volume density, O_2_ consumption rate per mitochondrion (i.e., normalized for) is reduced; (**E**) overall ROS formation measured using MitoSox; (**F**) however, when normalized for mitochondrial volume density, this increase in ROS is mitigated. (* denotes a significant difference between TNFα treated and untreated control groups; *p* < 0.05; *t*-test; cells were dissociated from ASM samples from *n* = 5 patients and divided into TNFα treated and untreated control groups). All data are presented as scatter plots with the lines indicating the mean values. These figures are modified from presentations of previously reported results [4].

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
