# Peer review of "Inflammation-Induced Protein Unfolding in Airway Smooth Muscle Triggers a Homeostatic Response in Mitochondria"

_ijms, 2020, doi:10.3390/ijms22010363_

Round 1

Reviewer 1 Report

This review manuscript puts forward and tries to argue for the following hypothesis: acute airway inflammation increases force generation and thus mitochondrial ATP production/oxygen consumption in airway smooth muscle cells. As this increases mitochondrial ROS production, a complex response involving a partial ER stress response and downstream regulatory mechanisms increasing mitochondria density balances the damaging effects by distributing respiratory ATP production on more mitochondria.

A large part of the evidence is based on two very recent and a number of previous studies of the author research group and it is discussed, mostly supported, by observations in the literature. The abstract does not clearly state that this is a hypothesis based on recent work and that this review attempts to put the pieces of the puzzle together to a consistent hypothesis. The sentence, lines 59 to 62 in the introduction would be better placed at an early position in the abstract to make this clear. Sentence 2 of the abstract (We found that 24 hours after….) gives the impression that this is an experimental study.

Much of the argumentation is based on indirect connections filled in by speculations and evidence for the causative nature of responses at the different levels and the order of events is poor. The mechanism how/reason why ROS from the mitochondria specifically activates one out of the 3 ER stress responses is unclear. Altogether, the text must be made more concise and own and literature observations and evidence more critically scrutinized.

Abstract:

The following two sentences in the abstract appear redundant:

This pathway (the pIRE1α/XBP1s ER stress pathway) initiates a homeostatic ER stress response directed towards increasing mitochondrial biogenesis and mitochondrial volume density to reduce O2 consumption and ROS formation.

TNFα promotes an increase in PGC1α expression (according to Figure 1 as a target of the pIRE1α/XBP1s ER stress pathway) and triggers mitochondrial biogenesis and an increase in mitochondrial volume density in ASM cells.

Some general questions regarding the arguments; Mitochondria (ROS, biogenesis, volume density, fission, mitophagy, mtUPR):

Distributing respiratory ATP production would not necessarily reduce ROS for the cell as the sum of leaked ROS from the respiratory chain is the same. Do the authors claim that the per cell increased superoxide distributed on a larger volume of mitochondria would be more efficiently detoxified by SODs and hydrogenperoxide detoxifying enzymes?

How can the mitochondrial density increase if fission and autophagy are increased at the same time? This would require a very much increased turnover of mitochondria, i.e. removal of fissioned bad mitochondria by autophagy and generation of new/extension of existing ‘good’ mitochondria by strongly increased synthesis of mitochondrial proteins, mt DNA and membrane lipids.

The effects of increased ROS on mitochondria is not discussed at all. It would be expected to result in misfolding and damage of mitochondrial proteins mounting a mtUPR.

Results:

The two sentences in lines 104 to 106 say the same, one positive (only) the other negative (not PERK ND ATF6 pathways).

Reference lacking: Line 136 ff: In previous studies, ….

Figures 1 and 2 should state that this are models.

Figures 3-6: It should be clearly stated that these figures are reprinted from previous publications. Not just a reference at the end.

The paragraph on lines 164 to 172 compares literature data to own data, but without arguing why, prefers the own data ‘clearly indicate’.

Line 188_ What exactly is meant with ‘these two energy sensing pathways’?

Some syntax errors:

Lines 125-126: In other cell types, previous studies reported that PINK1 mediates phosphorylate (phosphorylation) of Mfn2, which…

Lines 277-278: ‘However till date, no study has (been) explored the impact of ER stress and…’

Lines 289-291: ‘Considering all the facts, further illustrating studies are needed to understand the impact of inflammation in ER stress and mitochondrial crosstalk in asthma and also to find some novel therapeutic target(s).’ In these times, and in general, one should be careful with the term facts for experimental evidence/observations. But the ‘more studies are needed’ is definitely advisable.

Lines 282-284: Odd sentence. ‘Additionally, the effect of the  proinflammatory  cytokines other than TNFα whether has  any  impact  in  asthma  is  also a question to be answered.’

Author Response

Comment: This review manuscript puts forward and tries to argue for the following hypothesis: acute airway inflammation increases force generation and thus mitochondrial ATP production/oxygen consumption in airway smooth muscle cells. As this increases mitochondrial ROS production, a complex response involving a partial ER stress response and downstream regulatory mechanisms increasing mitochondria density balances the damaging effects by distributing respiratory ATP production on more mitochondria. A large part of the evidence is based on two very recent and a number of previous studies of the author research group and it is discussed, mostly supported, by observations in the literature. The abstract does not clearly state that this is a hypothesis based on recent work and that this review attempts to put the pieces of the puzzle together to a consistent hypothesis.

Response: We have revised the abstract to more clearly state that our review reflects a unifying hypothesis (conceptual framework) regarding the homeostatic role of ER stress in airway smooth muscle, based on our work and that of others.

Comment: The sentence, lines 59 to 62 in the introduction would be better placed at an early position in the abstract to make this clear.

Response: This sentence has been repositioned as recommended.

Comment: Sentence 2 of the abstract (We found that 24 hours after….) gives the impression that this is an experimental study.

Response: Here and elsewhere, we more clearly indicate that this is a review article that does not report original research.

Comment: Much of the argumentation is based on indirect connections filled in by speculations and evidence for the causative nature of responses at the different levels and the order of events is poor. The mechanism how/reason why ROS from the mitochondria specifically activates one out of the 3 ER stress responses is unclear. Altogether, the text must be made more concise and own and literature observations and evidence more critically scrutinized.

Response: We appreciate the reviewer’s concern and more clearly state that we are presenting a unifying hypothesis, which by nature is speculative. As with any hypothesis, cause and effect is difficult if not impossible to prove. However, results from our previous work and that of others is consistent with our hypothesis. Per the recommendation of the reviewer, we have revised the text to be more critical with respect to the evidence for cause and effect relationships of previous findings. The mechanism underlying the selective activation of pIRE1α/XBP1s ER stress pathway following TNFa exposure is unclear but could relate to the affinity of IRE1a in ASM cells to the Grp78/BiP protein, which triggers autophosphorylation of IRE1a and activation of the pIRE1α/XBP1s ER stress pathway. Selective activation of ER stress pathways has been reported in a number of previous studies in other cell types. Selective activation is not restricted to the pIRE1α/XBP1s ER stress pathway but can involve PERK or ATF6 pathways. Regardless of the source, the role of ROS in ASM cells is likely restricted to protein damage and unfolding rather than direct activation of ER stress pathways.

Comment: Abstract: The following two sentences in the abstract appear redundant:

This pathway (the pIRE1α/XBP1s ER stress pathway) initiates a homeostatic ER stress response directed towards increasing mitochondrial biogenesis and mitochondrial volume density to reduce O2 consumption and ROS formation.

TNFα promotes an increase in PGC1α expression (according to Figure 1 as a target of the pIRE1α/XBP1s ER stress pathway) and triggers mitochondrial biogenesis and an increase in mitochondrial volume density in ASM cells.

Response: We have revised these sentences in the abstract.

Comment: Some general questions regarding the arguments; Mitochondria (ROS, biogenesis, volume density, fission, mitophagy, mtUPR):

Distributing respiratory ATP production would not necessarily reduce ROS for the cell as the sum of leaked ROS from the respiratory chain is the same. Do the authors claim that the per cell increased superoxide distributed on a larger volume of mitochondria would be more efficiently detoxified by SODs and hydrogenperoxide detoxifying enzymes?

Response: The reviewer raises an important point. With an increase in mitochondrial volume, O2 consumption and ROS production per mitochondrion will be reduced, more distributed, and overall scavenging (by SOD, etc.) would be more effective. We now more clearly state this concept.

Comment: How can the mitochondrial density increase if fission and autophagy are increased at the same time? This would require a very much increased turnover of mitochondria, i.e. removal of fissioned bad mitochondria by autophagy and generation of new/extension of existing ‘good’ mitochondria by strongly increased synthesis of mitochondrial proteins, mt DNA and membrane lipids.

Response: We hypothesize that mitochondrial remodeling and quality control incudes: 1) mitochondrial fission and removal of damaged (“bad”) less effective (greater ROS production relative to O2 consumption) mitochondria via mitophagy, and 2) mitochondrial biogenesis resulting in higher quality, more effective mitochondria. This is an important part of our conceptual framework so we have revised the text to make this clearer.

Comment: The effects of increased ROS on mitochondria is not discussed at all. It would be expected to result in misfolding and damage of mitochondrial proteins mounting a mtUPR.

Response: The reviewer raises an important point that was implicit in our conceptual framework but was not clearly stated. We now include discussion of the mtUPR and coordination with the ER stress response.

Comment: Results: The two sentences in lines 104 to 106 say the same, one positive (only) the other negative (not PERK ND ATF6 pathways).

Response: In line 104, the positive control effect of tunicamycin in activating all three ER stress pathways (PERK, ATF6a and IRE1a) in ASM was mentioned. In the second sentence, we emphasize that TNFα selectively activates the pIRE1α/XBP1s ER stress pathway without having any significant impact on the other two ER stress pathways.

Comment: Reference lacking: Line 136 ff: In previous studies, ….

Response: We now include and appropriate reference.

Comment: Figures 1 and 2 should state that this are models.

Response: We now more clearly state that these figures reflect a conceptual model.

Comment: Figures 3-6: It should be clearly stated that these figures are reprinted from previous publications. Not just a reference at the end.

Response: We now more clearly state that these figures reflect modified presentation of previously reported results. The figures themselves were not previously included in the cited references.

Comment: The paragraph on lines 164 to 172 compares literature data to own data, but without arguing why, prefers the own data ‘clearly indicate’.

Response: This paragraph was rewritten to clarify.

Comment: Line 188_ What exactly is meant with ‘these two energy sensing pathways’?

Response: We revised this sentence to clarify what was meant.

Comment: Lines 125-126: In other cell types, previous studies reported that PINK1 mediates phosphorylate (phosphorylation) of Mfn2, which…

Response: This sentence was rewritten.

Comment: Lines 277-278: ‘However till date, no study has (been) explored the impact of ER stress and…’

Response: This sentence was rewritten.

Comment: Lines 289-291: ‘Considering all the facts, further illustrating studies are needed to understand the impact of inflammation in ER stress and mitochondrial crosstalk in asthma and also to find some novel therapeutic target(s).’ In these times, and in general, one should be careful with the term facts for experimental evidence/observations. But the ‘more studies are needed’ is definitely advisable.

Response: We agree, and this sentence was rewritten.

Comment: Lines 282-284: Odd sentence. ‘Additionally, the effect of the  proinflammatory  cytokines other than TNFα whether has  any  impact  in  asthma  is  also a question to be answered.’

Response: This sentence was rewritten.

Reviewer 2 Report

Reviewer Comments:

It is a well written review. Authors critically discussed the role of TNFα-induced ER stress pathway in maintaining mitochondrial homeostasis in ASM cells. The TNFα-induced ER stress increases PGC1α expression that not only enhances mitochondrial biogenesis, but also leads to decreased mitofusin-2 expression through activation of the PINK1/Parkin-mediated mitophagy pathway. A decrease in mitofusin-2 content increases mitochondrial fragmentation and disruption of mitochondrial tethering to the ER that leads to decreased Ca2+ influx into mitochondria.

The reviewer only has some minor comments.

Figure 2. Please include an explanation of the PTP leak in figure legend.

Figure 3. “α” is missing after PGC1. Please specify the data expression (Mean ± SD or Mean ± SEM). In addition, there were only two groups in Panel A, B, and C. It is better to use “t-test” rather than ANOVA for statistical comparison.

Figure 4, 5, and 6 have similar problem in statistical method.

Line 105. Authors stated “TNFα selectively activates only the pIRE1α/XBP1s ER stress pathway.” Did TNFα exposure increase CHOP expression? An increase in CHOP expression may also contribute to cell death during TNFα-induced ER stress.

Author Response

Response to Reviewer #2’s Comments:

Comment: It is a well written review. Authors critically discussed the role of TNFα-induced ER stress pathway in maintaining mitochondrial homeostasis in ASM cells. The TNFα-induced ER stress increases PGC1α expression that not only enhances mitochondrial biogenesis, but also leads to decreased mitofusin-2 expression through activation of the PINK1/Parkin-mediated mitophagy pathway. A decrease in mitofusin-2 content increases mitochondrial fragmentation and disruption of mitochondrial tethering to the ER that leads to decreased Ca2+ influx into mitochondria.

Response: We thank the reviewer for the very positive comments.

Comment: Figure 2. Please include an explanation of the PTP leak in figure legend.

Response: We now include an explanation of the PTP leak in figure 2 and in the text.

Comment: Figure 3. “α” is missing after PGC1.

Response: The text has been corrected.

Comment: Please specify the data expression (Mean ± SD or Mean ± SEM).

Response: We now specify that the data are expressed as means ± SD.

Comment: In addition, there were only two groups in Panel A, B, and C. It is better to use “t-test” rather than ANOVA for statistical comparison.

Response: We report the statistical analysis used in the original papers reporting these results and it was actually t-test. We now modify the manuscript accordingly.

Comment: Figure 4, 5, and 6 have similar problem in statistical method.

Response: We report the statistical analysis used in the original papers reporting these results and it was actually t-test. We now modify the manuscript accordingly.

Comment: Line 105. Authors stated “TNFα selectively activates only the pIRE1α/XBP1s ER stress pathway.” Did TNFα exposure increase CHOP expression? An increase in CHOP expression may also contribute to cell death during TNFα-induced ER stress.

Response: We did measure CHOP expression in ASM cells and we found that it was unaffected by TNFa exposure. In addition, we found no evidence for increased ASM cell apoptosis. Activation of CHOP and cell apoptosis in response to ER stress is generally associated with the ATF-6 pathway and indirectly to PERK pathway, not the IRE1a pathway that is selectively activated in ASM cells in response to TNFa exposure. In contrast, tunicamycin exposure activated all three ER stress pathways, and was associated with substantial cell apoptosis. These ‘negative’ results regarding changes in CHOP and apoptosis following TNFa exposure are now included in the text.

Round 2

Reviewer 1 Report

Revision is fine. Thank you for addressing the comments appropriately. There remain just a few small things:

Line 33: ‘…reduced ROS formation by individual mitochondrion…’; I think ‘mitochondria’ would be the correct form. Please check. Same in lines 91 and 382.

Line 59: references (25,30) might already be used here at the first mention of the mtUPR.

Lines 376-377: ‘TNFα plays (a) major role in force generation and ATP consumption by ASM

during airway inflammation and induce(s) mitochondrial oxidative phosphorylation…’

Author Response

Comment: Revision is fine. Thank you for addressing the comments appropriately.

Response: We thank the reviewer for the very positive comments.

Comment: Line 33: ‘…reduced ROS formation by individual mitochondrion…’; I think ‘mitochondria’ would be the correct form. Please check. Same in lines 91 and 382.

Response: We hereby want to focus on single unit of mitochondria hence we use the singular form “mitochondrion”.

Comment: Line 59: references (25,30) might already be used here at the first mention of the mtUPR.

Response: The references are repositioned in the manuscript accordingly.

Comment: Lines 376-377: ‘TNFα plays (a) major role in force generation and ATP consumption by ASM during airway inflammation and induce(s) mitochondrial oxidative phosphorylation…’

Response: The sentence is revised in the manuscript.